# OpenReview forum: "Black-Box Adversarial Attacks on LLM-Based Code Completion"
_ICLR.cc/2025/Workshop/BuildingTrust — BuildingTrust_

### Official Review · Reviewer_kmJE · 2025-02-21
**Well written paper but inconsistent threat model**

**Rating:** 4
**Confidence:** 3

**Review:**

Pros:
- This paper presents an interesting innovation: inject a comment into the code that the user wants the LM to complete.  This comment is adversarially optimized so that it will cause the LM to insert a vulnerability into the code.
- The writing is very clear and the graphics are beautiful
- The experiments are fairly extensive and conducted against production models.

Cons:
- The threat model is very unclear in multiple ways:
a.  If the attacker is able to modify the user input into the LM, why can they not directly modify the output of the LM to insert a vulnerability?  This would be far simpler than trying to perform a prompt injection attack
b.  How is the attacker able to modify the user inputs at all?  It seems like these would probably be encrypted if they're being sent to a remote server hosting the LM.
c.  The attackers use something like a black-box coordinate descent attack, but wouldn't this significantly impact the speed of the code completion to the point where the user should notice that something is wrong.

I think that this would be a really strong paper if the threat model were clarified, but as it stands I don't understand how this attack is practical or realistic.

---

### Official Review · Reviewer_oCYz · 2025-02-26
**Evaluating INSEC: A Practical Black-Box Attack on LLM-Based Code Completion and Its Security Implications**

**Rating:** 7
**Confidence:** 3

**Review:**

### Summary:
The paper examines security vulnerabilities in LLM-based code completion tools, demonstrating that black-box models can be influenced to generate insecure code at a significantly higher rate. The authors introduce INSEC, an attack method that strategically inserts adversarial comments into code prompts, resulting in a 50% increase in insecure code suggestions while maintaining functional correctness. Unlike prior white-box attacks that require modifying model weights or training data, INSEC operates entirely in a black-box setting, making it highly practical and cost-effective, with a development cost of less than $10. The paper evaluates INSEC’s effectiveness on several leading models, including OpenAI’s API and GitHub Copilot, and further validates its real-world impact by implementing an IDE plugin that seamlessly injects the attack. These findings highlight the need for enhanced security measures in AI-assisted coding tools.

### Strengths:
1) **Novel Attack Strategy:** INSEC introduces a practical and stealthy black-box attack that demonstrates how LLM-based code completion can be manipulated without modifying model internals.
2) **Broad Empirical Validation:** The attack is rigorously evaluated on multiple open-source (StarCoder, CodeLlama) and commercial (GPT-3.5, GitHub Copilot) code completion engines. The effectiveness of INSEC across different models, programming languages, and CWEs strengthens its impact.
3)**Theoretical Foundation:** The authors provide well-structured mathematical formulations and an optimization approach to derive effective adversarial comment strings.

### Weaknesses:
1) **Limited Motivation for Attackers:** While the attack is effective, the paper does not fully justify why an adversary would go through the effort of biasing code-completion engines.
2) **Impact on Model Robustness:** While the paper ensures that INSEC maintains functional correctness in security-sensitive tasks, it does not evaluate whether the attack degrades model performance on general, non-security-related completions. Understanding if INSEC affects the broader utility of code completion models would be valuable.
3) **Minor Typos and Formatting Issues:** Some minor errors, such as “mathch” instead of “match,” should be corrected for clarity and professionalism.

---

### Official Review · Reviewer_veEJ · 2025-03-02

**Rating:** 5
**Confidence:** 4

**Review:**

This paper introduces a jailbreaking attack on code generation models that results in code with more security vulnerabilities. While the attack is successful and intuitive (injecting attacks as optimized strings in comments) and well ablated, they are not particularly surprising considering the large body of jailbreaking work. The attack itself seems like an extension of PAIR to this code setting, and insecure code generation should be a subset of harmful generation. It would be more interesting if the authors consider alternative and more realistic settings such as code agents.

---

### Decision · Program_Chairs · 2025-03-01

Accept